# Relational recurrent neural networks

**Adam Santoro\*$^{\alpha}$, Ryan Faulkner\*$^{\alpha}$, David Raposo\*$^{\alpha}$, Jack Rae$^{\alpha\beta}$, Mike Chrzanowski$^{\alpha}$, Théophane Weber$^{\alpha}$, Daan Wierstra$^{\alpha}$, Oriol Vinyals$^{\alpha}$, Razvan Pascanu$^{\alpha}$, Timothy Lillicrap$^{\alpha\beta}$**

$^{\alpha}$DeepMind
London, United Kingdom

$^{\beta}$CoMPLEX, Computer Science, University College London
London, United Kingdom

```
{adamsantoro; rfaulk; draposo; jwrae; chrzanowskim;
theophane; weirstra; vinyals; razp; countzero}@google.com
```

## Abstract

Memory-based neural networks model temporal data by leveraging an ability to remember information for long periods. It is unclear, however, whether they also have an ability to perform complex relational reasoning with the information they remember. Here, we first confirm our intuitions that standard memory architectures may struggle at tasks that heavily involve an understanding of the ways in which entities are connected – i.e., tasks involving relational reasoning. We then improve upon these deficits by using a new memory module – a *Relational Memory Core* (RMC) – which employs multi-head dot product attention to allow memories to interact. Finally, we test the RMC on a suite of tasks that may profit from more capable relational reasoning across sequential information, and show large gains in RL domains (e.g. Mini PacMan), program evaluation, and language modeling, achieving state-of-the-art results on the WikiText-103, Project Gutenberg, and GigaWord datasets.

## 1 Introduction

Humans use sophisticated memory systems to access and reason about important information regardless of when it was initially perceived [1, 2]. In neural network research many successful approaches to modeling sequential data also use memory systems, such as LSTMs [3] and memory-augmented neural networks generally [4–7]. Bolstered by augmented memory capacities, bounded computational costs over time, and an ability to deal with vanishing gradients, these networks learn to correlate events across time to be proficient at *storing* and *retrieving* information.

Here we propose that it is fruitful to consider *memory interactions* along with storage and retrieval. Although current models can learn to compartmentalize and relate distributed, vectorized memories, they are not biased towards doing so explicitly. We hypothesize that such a bias may allow a model to better understand how memories are related, and hence may give it a better capacity for relational reasoning over time. We begin by demonstrating that current models do indeed struggle in this domain by developing a toy task to stress relational reasoning of sequential information. Using a new *Relational Memory Core* (RMC), which uses multi-head dot product attention to allow memories to interact with each other, we solve and analyze this toy problem. We then apply the RMC to a suite of tasks that may profit from more explicit memory-memory interactions, and hence, a potentially

increased capacity for relational reasoning across time: partially observed reinforcement learning tasks, program evaluation, and language modeling on the Wikitext-103, Project Gutenberg, and GigaWord datasets.

## 2 Relational reasoning

We take relational reasoning to be the process of understanding the ways in which entities are connected and using this understanding to accomplish some higher order goal [8]. For example, consider sorting the distances of various trees to a park bench: the *relations* (distances) between the *entities* (trees and bench) are compared and contrasted to produce the solution, which could not be reached if one reasoned about the properties (positions) of each individual entity in isolation.

Since we can often quite fluidly define what constitutes an "entity" or a "relation", one can imagine a spectrum of neural network inductive biases that can be cast in the language of relational reasoning [1]. For example, a convolutional kernel can be said to compute a relation (linear combination) of the entities (pixels) within a receptive field. Some previous approaches make the relational inductive bias more explicit: in message passing neural networks [e.g. 9–12], the nodes comprise the entities and relations are computed using learnable functions applied to nodes connected with an edge, or sometimes reducing the relational function to a weighted sum of the source entities [e.g. 13, 14]. In Relation Networks [15–17] entities are obtained by exploiting spatial locality in the input image, and the model focuses on computing binary relations between each entity pair. Even further, some approaches emphasize that more capable reasoning may be possible by employing simple computational principles; by recognizing that relations might not always be tied to proximity in space, *non-local* computations may be better able to capture the relations between entities located far away from each other [18, 19].

In the temporal domain relational reasoning could comprise a capacity to compare and contrast information seen at different points in time [20]. Here, attention mechanisms [e.g. 21, 22] implicitly perform some form of relational reasoning; if previous hidden states are interpreted as entities, then computing a weighted sum of entities using attention helps to remove the *locality bias* present in vanilla RNNs, allowing embeddings to be better related using content rather than proximity.

Since our current architectures solve complicated temporal tasks they must have some capacity for temporal relational reasoning. However, it is unclear whether their inductive biases are limiting, and whether these limitations can be exposed with tasks demanding particular types of temporal relational reasoning. For example, memory-augmented neural networks [4–7] solve a compartmentalization problem with a slot-based memory matrix, but may have a harder time allowing memories to interact, or relate, with one another once they are encoded. LSTMs [3, 23], on the other hand, pack all information into a common hidden memory vector, potentially making compartmentalization and relational reasoning more difficult.

## 3 Model

Our guiding design principle is to provide an architectural backbone upon which a model can learn to compartmentalize information, and learn to compute interactions between compartmentalized information. To accomplish this we assemble building blocks from LSTMs, memory-augmented neural networks, and non-local networks (in particular, the Transformer seq2seq model [22]). Similar to memory-augmented architectures we consider a fixed set of memory slots; however, we allow for interactions *between* memory slots using an attention mechanism. As we will describe, in contrast to previous work we apply attention between memories at a single time step, and not across all previous representations computed from all previous observations.

### 3.1 Allowing memories to interact using multi-head dot product attention

We will first assume that we do not need to consider memory encoding; that is, that we already have some stored memories in matrix $M$, with row-wise compartmentalized memories $m_i$. To allow memories to interact we employ *multi-head dot product attention* (MHDPA) [22], also known as

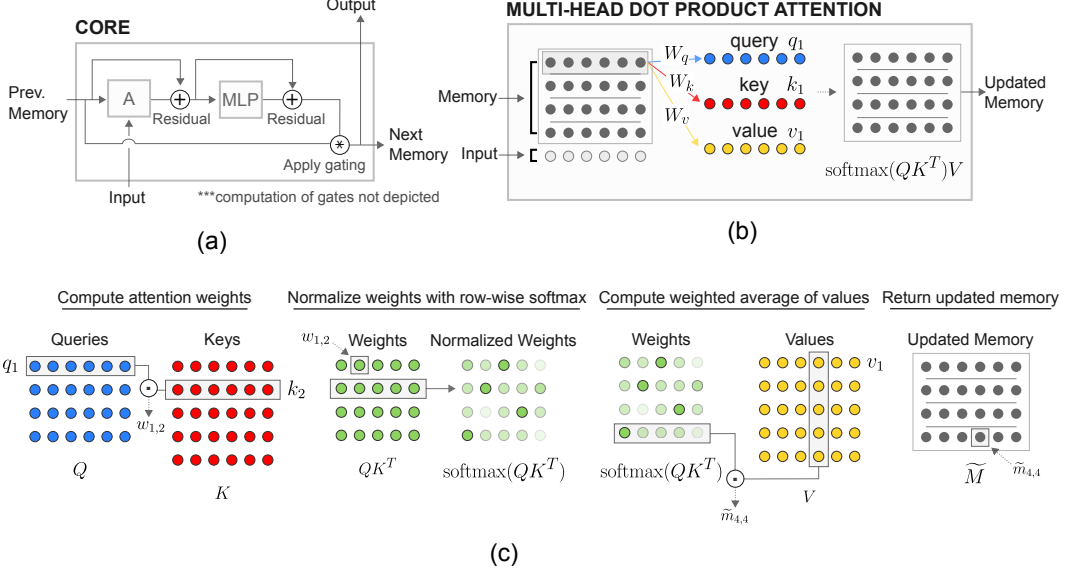

Figure 1: **Relational Memory Core**. (a) The RMC receives a previous memory matrix and input vector as inputs, which are passed to the MHDPA module labeled with an "A". (b). Linear projections are computed for each memory slot, and input vector, using row-wise shared weights $W^q$ for the queries, $W^k$ for the keys, and $W^v$ for the values. (c) The queries, keys, and values are then compiled into matrices and $\text{softmax}(QK^T)V$ is computed. The output of this computation is a new memory where information is blended across memories based on their attention weights. An MLP is applied row-wise to the output of the MHDPA module (a), and the resultant memory matrix is gated, and passed on as the core output or next memory state.

*self-attention.* Using MHDPA, each memory will attend over all of the other memories, and will update its content based on the attended information.

First, a simple linear projection is used to construct queries ($Q = MW^q$), keys ($K = MW^k$), and values ($V = MW^v$) for each memory (i.e. row $m_i$) in matrix $M$. Next, we use the queries, $Q$, to perform a scaled dot-product attention over the keys, $K$. The returned scalars can be put through a softmax-function to produce a set of weights, which can then be used to return a weighted average of values from $V$ as $A(Q, K, V) = \text{softmax}\left(\frac{QK^T}{\sqrt{d_k}}\right)V$, where $d_k$ is the dimensionality of the key vectors used as a scaling factor. Equivalently:

$$A_\theta(M) = \text{softmax}\left(\frac{MW^q(MW^k)^T}{\sqrt{d_k}}\right)MW^v, \text{ where } \theta = (W^q, W^k, W^v) \qquad (1)$$

The output of $A_\theta(M)$, which we will denote as $\widetilde{M}$, is a matrix with the same dimensionality as $M$. $\widetilde{M}$ can be interpreted as a proposed update to $M$, with each $\widetilde{m}_i$ comprising information from memories $m_j$. Thus, in one step of attention each memory is updated with information originating from other memories, and it is up to the model to learn (via parameters $W^q$, $W^k$, and $W^v$) how to shuttle information from memory to memory.

As implied by the name, MHDPA uses multiple heads. We implement this producing $h$ sets of queries, keys, and values, using unique parameters to compute a linear projection from the original memory for each head $h$. We then independently apply an attention operation for each head. For example, if $M$ is an $N \times F$ dimensional matrix and we employ two attention heads, then we compute $\widetilde{M^1} = A_\theta(M)$ and $\widetilde{M^2} = A_\phi(M)$, where $\widetilde{M^1}$ and $\widetilde{M^2}$ are $N \times F/2$ matrices, $\theta$ and $\phi$ denote unique parameters for the linear projections to produce the queries, keys, and values, and $\widetilde{M} = [\widetilde{M^1} : \widetilde{M^2}]$, where $[:]$ denotes column-wise concatenation. Intuitively, heads could be useful for letting a memory share different information, to different targets, using each head.

## 3.2 Encoding new memories

We assumed that we already had a matrix of memories $M$. Of course, memories instead need to be encoded as new inputs are received. Suppose then that $M$ is some randomly initialised memory. We can efficiently incorporate new information $x$ into $M$ with a simple modification to equation 1:

$$\widetilde{M} = \text{softmax}\left(\frac{MW^q([M;x]W^k)^T}{\sqrt{d^k}}\right)[M;x]W^v, \tag{2}$$

where we use $[M;x]$ to denote the row-wise concatenation of $M$ and $x$. Since we use $[M;x]$ when computing the keys and values, and only $M$ when computing the queries, $\widetilde{M}$ is a matrix with same dimensionality as $M$. Thus, equation 2 is a memory-size preserving attention operation that includes attention over the memories and the new observations. Notably, we use the same attention operation to efficiently compute memory interactions and to incorporate new information.

We also note the possible utility of this operation when the memory consists of a single vector rather than a matrix. In this case the model may learn to pick and choose which information from the input should be written into the vector memory state by learning how to attend to the input, conditioned on what is contained in the memory already. This is possible in LSTMs via the gates, though at a different granularity. We return to this idea, and the possible compartmentalization that can occur via the heads even in the single-memory-slot case, in the discussion.

## 3.3 Introducing recurrence and embedding into an LSTM

Suppose we have a temporal dimension with new observations at each timestep, $x_t$. Since $M$ and $\widetilde{M}$ are the same dimensionality, we can naively introduce recurrence by first randomly initialising $M$, and then updating it with $\widetilde{M}$ at each timestep. We chose to do this by embedding this update into an LSTM. Suppose memory matrix $M$ can be interpreted as a matrix of cell states, usually denoted as $C$, for a 2-dimensional LSTM. We can make the operations of individual memories $m_i$ nearly identical to those in a normal LSTM cell state as follows (subscripts are overloaded to denote the row from a matrix, and timestep; e.g., $m_{i,t}$ is the $i^{th}$ row from $M$ at time $t$).

$$s_{i,t} = (h_{i,t-1}, m_{i,t-1}) \tag{3}$$

$$f_{i,t} = W^f x_t + U^f h_{i,t-1} + b^f \tag{4}$$

$$i_{i,t} = W^i x_t + U^i h_{i,t-1} + b^i \tag{5}$$

$$o_{i,t} = W^o x_t + U^o h_{i,t-1} + b^o \tag{6}$$

$$m_{i,t} = \sigma(f_{i,t} + \tilde{b}^f) \circ m_{i,t-1} + \sigma(i_{i,t}) \circ \underbrace{g_\psi(\widetilde{m}_{i,t})} \tag{7}$$

$$h_{i,t} = \sigma(o_{i,t}) \circ \tanh(m_{i,t}) \tag{8}$$

$$s_{i,t+1} = (m_{i,t}, h_{i,t}) \tag{9}$$

The underbrace denotes the modification to a standard LSTM. In practice we did not find output gates necessary – please see the url in the footnote for our Tensorflow implementation of this model in the Sonnet library [2], and for the exact formulation we used, including our choice for the $g_\psi$ function (briefly, we found a row/memory-wise MLP with layer normalisation to work best). There is also an interesting opportunity to introduce a different kind of gating, which we call 'memory' gating, which resembles previous gating ideas [24, 3]. Instead of producing scalar gates for each individual unit ('unit' gating), we can produce scalar gates for each memory row by converting $W^f$, $W^i$, $W^o$, $U^f$, $U^i$, and $U^o$ from weight matrices into weight vectors, and by replacing the element-wise product in the gating equations with scalar-vector multiplication.

Since parameters $W^f$, $W^i$, $W^o$, $U^f$, $U^i$, $U^o$, and $\psi$ are shared for each $m_i$, we can modify the number of memories without affecting the number of parameters. Thus, tuning the number of memories and the size of each memory can be used to balance the overall storage capacity (equal to the total number of units, or elements, in $M$) and the number of parameters (proportional to the dimensionality of $m_i$). We find in our experiments that some tasks require more, but not necessarily larger, memories, and others such as language modeling require fewer, larger memories.

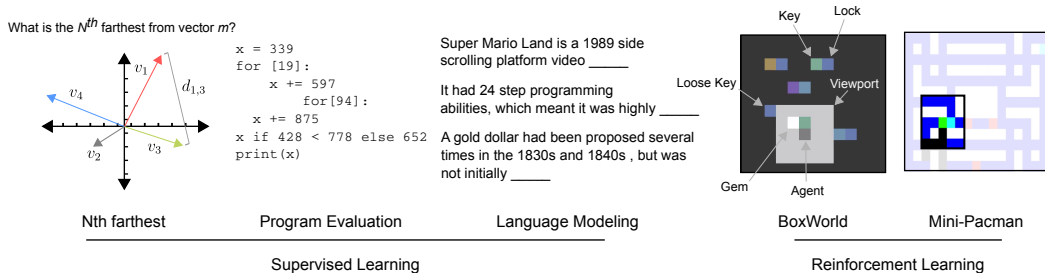

Figure 2: **Tasks**. We tested the RMC on a suite of supervised and reinforcement learning tasks. Notable are the $N^{th}$ Farthest toy task and language modeling. In the former, the solution requires explicit relational reasoning since the model must sort *distance relations* between vectors, and not the vectors themselves. The latter tests the model on a large quantity of natural data and allows us to compare performance to well-tuned models.

Thus, we have a number of tune-able parameters: the number of memories, the size of each memory, the number of attention heads, the number of steps of attention, the gating method, and the post-attention processor $g_\psi$. In the appendix we list the exact configurations for each task.

## 4 Experiments

Here we briefly outline the tasks on which we applied the RMC, and direct the reader to the appendix for full details on each task and details on hyperparameter settings for the model.

### 4.1 Illustrative supervised tasks

$N^{th}$ **Farthest** The $N^{th}$ Farthest task is designed to stress a capacity for relational reasoning across time. Inputs are a sequence of randomly sampled vectors, and targets are answers to a question of the form: "What is the $n^{th}$ farthest vector (in Euclidean distance) from vector $m$?", where the vector values, their IDs, $n$, and $m$ are randomly sampled per sequence. It is not enough to simply encode and retrieve information as in a copy task. Instead, a model must compute all pairwise distance relations to the reference vector $m$, which might also lie in memory, or might not have even been provided as input yet. It must then implicitly sort these distances to produce the answer. We emphasize that the model must sort *distance relations* between vectors, and not the vectors themselves.

**Program Evaluation** The *Learning to Execute* (**LTE**) dataset [25] consists of algorithmic snippets from a Turing complete programming language of pseudo-code, and is broken down into three categories: *addition*, *control*, and *full program*. Inputs are a sequence of characters over an alphanumeric vocabulary representing such snippets, and the target is a numeric sequence of characters that is the execution output for the given programmatic input. Given that the snippets involve symbolic manipulation of variables, we felt it could strain a model's capacity for relational reasoning; since symbolic operators can be interpreted as defining a relation over the operands, successful learning could reflect an understanding of this relation. To also assess model performance on classical sequence tasks we also evaluated on *memorization tasks*, in which the output is simply a permuted form of the input rather than an evaluation from a set of operational instructions. See the appendix for further experimental details.

### 4.2 Reinforcement learning

**Mini Pacman with viewport** We follow the formulation of Mini Pacman from [26]. Briefly, the agent navigates a maze to collect food while being chased by ghosts. However, we implement this task with a viewport: a $5 \times 5$ window surrounding the agent that comprises the perceptual input. The task is therefore partially observable, since the agent must navigate the space and take in information through this viewport. Thus, the agent must predict the dynamics of the ghosts *in memory*, and plan its navigation accordingly, also based on remembered information about which food has already been

picked up. We also point the reader to the appendix for a description and results of another RL task called BoxWorld, which demands relational reasoning in memory space.

### 4.3 Language Modeling

Finally, we investigate the task of word-based language modeling. We model the conditional probability $p(w_t|w_{<t})$ of a word $w_t$ given a sequence of observed words $w_{<t} = (w_{t-1}, w_{t-2}, \ldots, w_1)$. Language models can be directly applied to predictive keyboard and search-phrase completion, or they can be used as components within larger systems, e.g. machine translation [27], speech recognition [28], and information retrieval [29]. RNNs, and most notably LSTMs, have proven to be state-of-the-art on many competitive language modeling benchmarks such as Penn Treebank [30, 31], WikiText-103 [32, 33], and the One Billion Word Benchmark [34, 35]. As a sequential reasoning task, language modeling allows us to assess the RMC's ability to process information over time on a large quantity of natural data, and compare it to well-tuned models.

We focus on datasets with contiguous sentences and a moderately large amount of data. WikiText-103 satisfies this set of requirements as it consists of Wikipedia articles shuffled at the article level with roughly $100M$ training tokens, as do two stylistically different sources of text data: books from Project Gutenberg[3] and news articles from GigaWord v5 [36]. Using the same processing from [32] these datasets consist of $180M$ training tokens and $4B$ training tokens respectively, thus they cover a range of styles and corpus sizes. We choose a similar vocabulary size for all three datasets of approximately $250,000$, which is large enough to include rare words and numeric values.

## 5 Results

### 5.1 $N^{th}$ Farthest

This task revealed a stark difference between our LSTM and DNC baselines and RMC when training on 16-dimensional vector inputs. Both LSTM and DNC models failing to surpass $30\%$ best batch accuracy and the RMC consistently achieving $91\%$ at the end of training (see figure 5 in the appendix for training curves). The RMC achieved similar performance when the difficulty of the task was increased by using 32-dimensional vectors, placing a greater demand on high-fidelity memory storage. However, this performance was less robust with only a small number of seeds/model configurations demonstrating this performance, in contrast to the 16-dimensional vector case where most model configurations succeeded.

An attention analysis revealed some notable features of the RMC's internal functions. Figure 3 shows attention weights in the RMC's memory throughout a sequence: the first row contains a sequence where the reference vector $m$ was observed last; in the second row it was observed first; and in the last row it was observed in the middle of the sequence. Before $m$ is seen the model seems to shuttle input information into one or two memory slots, as shown by the high attention weights from these slots' queries to the input key. After $m$ is seen, most evident in row three of the figure, the model tends to change its attention behaviour, with all the memory slots preferentially focusing attention on those particular memories to which the $m$ was written. Although this attention analysis provides some useful insights, the conclusions we can make are limited since even after a single round of attention the memory can become highly distributed, making any interpretations about information compartmentalisation potentially inaccurate.

### 5.2 Program Evaluation

Program evaluation performance was assessed via the *Learning to Execute* tasks [25]. We evaluated a number of baselines alongside the RMC including an LSTM [3, 37], DNC [5], and a bank of LSTMs resembling Recurrent Entity Networks [38] (EntNet) - the configurations for each of these is described in the appendix. Best test batch accuracy results are shown in Table 1. The RMC performs at least as well as all of the baselines on each task. It is marginally surpassed by a small fraction of performance on the *double* memorization task, but both models effectively solve this task. Further, the results of the RMC outperform all equivalent tasks from [25] which use teacher forcing even when evaluating model performance. It's worth noting that we observed better results when we trained in a

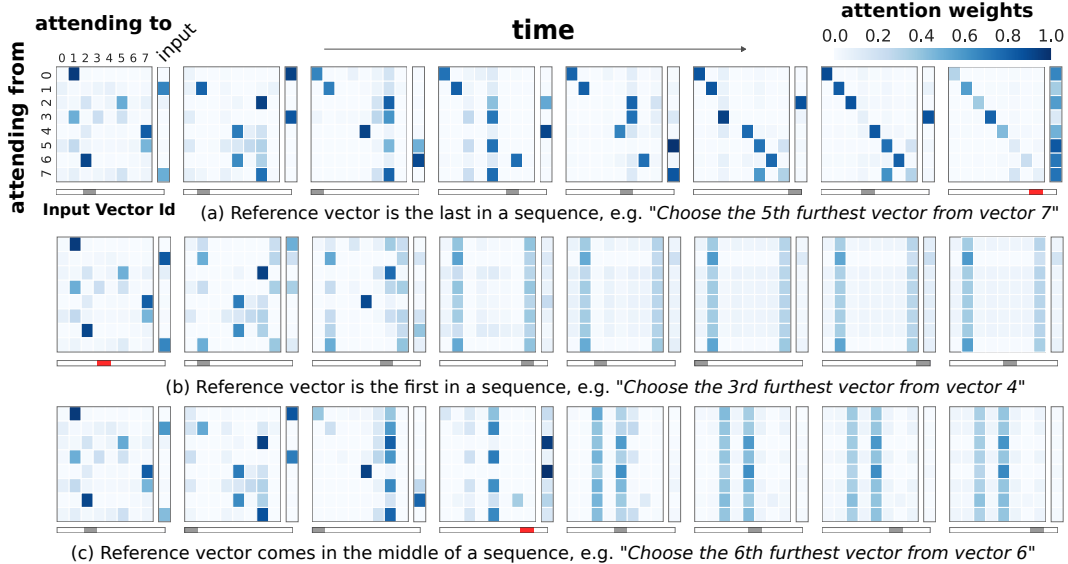

(a) Reference vector is the last in a sequence, e.g. "Choose the 5th furthest vector from vector 7"

(b) Reference vector is the first in a sequence, e.g. "Choose the 3rd furthest vector from vector 4"

(c) Reference vector comes in the middle of a sequence, e.g. "Choose the 6th furthest vector from vector 6"

Figure 3: **Model analysis**. Each row depicts the attention matrix at each timestep of a particular sequence. The text beneath spells out the particular task for the sequence, which was encoded and provided to the model as an input. We mark in red the vector that is referenced in the task: e.g., if the model is to choose the $2^{nd}$ farthest vector from vector 7, then the time point at which vector 7 was input to the model is depicted in red. A single attention matrix shows the attention weights from one particular memory slot (y-axis) to another memory slot (columns), or the input (offset column), with the numbers denoting the memory slot and "input" denoting the input embedding.

non-auto-regressive fashion - that is, with no teacher forcing during training. This is likely related to the effect that relaxing the ground truth requirement has on improving model generalization [39] and hence, performance. It is perhaps more pronounced in these tasks due to the independence of output token probabilities and also the sharply uni-modal nature of the output distribution (that is, there is no ambiguity in the answer given the program).

Table 1: Test per character Accuracy on Program Evaluation and Memorization tasks.

| Model | Add | Control | Program | Copy | Reverse | Double |
|---|---|---|---|---|---|---|
| LSTM [3, 37] | 99.8 | 97.4 | 66.1 | 99.8 | 99.7 | 99.7 |
| EntNet [38] | 98.4 | 98.0 | 73.4 | 91.8 | **100.0** | 62.3 |
| DNC [5] | 99.4 | 83.8 | 69.5 | **100.0** | **100.0** | **100.0** |
| Relational Memory Core | **99.9** | **99.6** | **79.0** | **100.0** | **100.0** | 99.8 |

Table 2: Validation and test perplexities on WikiText-103, Project Gutenberg, and GigaWord v5.

| | WikiText-103 | | Gutenberg | | GigaWord |
|---|---|---|---|---|---|
| | Valid. | Test | Valid | Test | Test |
| LSTM [40] | - | 48.7 | - | - | - |
| Temporal CNN [41] | - | 45.2 | - | - | - |
| Gated CNN [42] | - | 37.2 | - | - | - |
| LSTM [32] | 34.1 | 34.3 | 41.8 | 45.5 | 43.7 |
| Quasi-RNN [43] | 32 | 33 | - | - | - |
| Relational Memory Core | **30.8** | **31.6** | **39.2** | **42.0** | **38.3** |

### 5.3 Mini-Pacman

In Mini Pacman with viewport the RMC achieved approximately 100 points more than an LSTM (677 vs. 550), and when trained with the full observation the RMC nearly doubled the performance of an LSTM (1159 vs. 598, figure 10).

### 5.4 Language Modeling

For all three language modeling tasks we observe lower perplexity when using the relational memory core, with a drop of $1.4 - 5.4$ perplexity over the best published results. Although small, this constitutes a $5 - 12\%$ relative improvement and appears to be consistent across tasks of varying size and style. For WikiText-103, we see this can be compared to LSTM architectures [5, 32], convolutional models [42] and hybrid recurrent-convolutional models [43].

The model learns with a slightly better data efficiency than an LSTM (appendix figure 11). The RMC scored highly when the number of context words provided during evaluation were relatively few, compared to an LSTM which profited much more from a larger context (supplementary figure 12). This could be because RMC better captures short-term relations, and hence only needs a relatively small context for accurate modeling. Inspecting the perplexity broken down by word frequency in supplementary table 3, we see the RMC improved the modeling of frequent words, and this is where the drop in overall perplexity is obtained.

## 6 Discussion

A number of other approaches have shown success in modeling sequential information by using a growing buffer of previous states [21, 22]. These models better capture long-distance interactions, since their computations are not biased by temporally local proximity. However, there are serious scaling issues for these models when the number of timesteps is large, or even unbounded, such as in online reinforcement learning (e.g., in the real world). Thus, some decisions need to be made regarding the size of the past-embedding buffer that should be stored, whether it should be a rolling window, how computations should be cached and propagated across time, etc. These considerations make it difficult to directly compare these approaches in these online settings. Nonetheless, we believe that a blend of purely recurrent approaches with those that scale with time could be a fruitful pursuit: perhaps the model accumulates memories losslessly for some chunk of time, then learns to compress it in a recurrent core before moving onto processing a subsequent chunk.

We believe it is difficult to agree on a definition for "relational reasoning", and prefer the intuition that "relational reasoning" describes an ability or capacity to understand the ways in which entities are related. Since it is a capacity, it must be measured behaviourally, using tasks that we know to demand reasoning about the ways in which entities are related. Thus, since we cannot formally prove that some series of computations will necessarily result in improved relational reasoning, we must rely on empirical verification that some computations may or may not be correlated with improved relational reasoning. We hypothesized that memory-memory interactions may be underlie such computations, and proposed intuitions for the specific memory mechanisms that may better equip a model for complex relational reasoning. Namely, by explicitly allowing memories to interact either with each other, with the input, or both via MHDPA, we demonstrated improved performance on tasks demanding relational reasoning across time. We would like to emphasize, however, that while these intuitions guided our design of the model, and while the analysis of the model in the $N^{th}$ farthest task aligned with our intuitions, we cannot necessarily make any concrete claims as to the causal influence of our design choices on the model's capacity for relational reasoning, or as to the computations taking place within the model and how they may map to traditional approaches for thinking about relational reasoning. Thus, we consider our results primarily as evidence of *improved function* – if a model can better solve tasks that require relational reasoning, then it must have an increased capacity for relational reasoning, even if we do not precisely know why it may have this increased capacity. In this light the RMC may be usefully viewed from multiple vantages, and these vantages may offer ideas for further improvements.

Our model has multiple mechanisms for forming and allowing for interactions between memory vectors: slicing the memory matrix row-wise into slots, and column-wise into heads. Each has its own advantages (computations on slots share parameters, while having more heads and a larger

memory size takes advantage of more parameters). We don't yet understand the interplay, but we note some empirical findings. First, in the the $N^{th}$ farthest task a model with a single memory slot performed better when it had more attention heads, though in all cases it performed worse than a model with many memory slots. Second, in language modeling, our model used a single memory slot. The reasons for choosing a single memory here were mainly due to the need for a large number of parameters for LM in general (hence the large size for the single memory slot), and the inability to quickly run a model with both a large number of parameters and multiple memory slots. Thus, we do not necessarily claim that a single memory slot is best for language modeling, rather, we emphasize an interesting trade-off between number of memories and individual memory size, which may be a task specific ratio that can be tuned. Moreover, in program evaluation, an intermediate solution worked well across subtasks (4 slots and heads), though some performed best with 1 memory, and others with 8.

Altogether, our results show that explicit modeling of memory interactions improves performance in a reinforcement learning task, alongside program evaluation, comparative reasoning, and language modeling, demonstrating the value of instilling a capacity for relational reasoning in recurrent neural networks.

## Acknowledgements

We thank Caglar Gulcehre, Matt Botvinick, Vinicius Zambaldi, Charles Blundell, Sébastien Racaniere, Chloe Hillier, Victoria Langston, and many others on the DeepMind team for critical feedback, discussions, and support.

## Footnotes

[1]Indeed, in the broadest sense any multivariable function must be considered "relational."

[2]https://github.com/deepmind/sonnet/blob/master/sonnet/python/modules/relational_memory.py

[3]Project Gutenberg. (n.d.). Retrieved January 2, 2018, from www.gutenberg.org

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
