[Supplementary Material]

# A Further task details, analyses, and model configurations

In the following sections we provide further details on the experiments and the model configurations. We will sometimes refer to the following terms when describing the model:

- "total units": The total number of elements in the memory matrix $M$. Equivalent to the size of each memory multiplied by the number of memories.

- "num heads": The number of attention heads; i.e., the number of unique sets of queries, keys, and values produced for the memories.

- "memory slots" or "number of memories": Equivalent to the number of rows in matrix $M$.

- "num blocks": The number of iterations of attention performed at each time-step.

- "gate style": Gating per unit or per memory slot

## A.1 $N^{th}$ Farthest

Inputs consisted of sequences of eight randomly sampled, 16-dimensional vectors from a uniform distribution $x_t \sim \mathcal{U}(-1, 1)$, and vector labels $l_t \sim \{1, 2, ..., 8\}$, encoded as a one-hot vectors and sampled without replacement. Labels were *sampled* and hence did not correspond to the time-points at which the vectors were presented to the model. Appended to each vector-label input was the task specification (i.e., the values of $n$ and $m$ for that sequence), also encoded as one-hot vectors. Thus, an input for time-step $t$ was a 40-dimensional vector $(x_t; l_t; n; m)$.

For all models (RMC, LSTM, DNC) we used the Adam optimiser [44] with a batch size of 1600, learning rates tuned between $1e^{-5}$ and $1e^{-3}$, and trained using a softmax cross entropy loss function. All the models had an equivalent 4-layer MLP (256 units per layer with ReLu non-linearities) to process their outputs to produce logits for the softmax. Learning rate did not seem to influence performance, so we settled on $1e^{-4}$ for the final experiments.

For the LSTM and DNC, architecture parameters seemingly made no difference to model performance. For the LSTM we tried hidden sizes ranging from 64 up to 4096 units, and for the DNC we tried 1, 8, or 16 memories, 128, 512, or 1024 memory sizes (which we tied to the controller LSTM size), and 1, 2, or 4 memory reads & writes. The DNC used a 2-layer LSTM controller.

For the RMC we used 1, 8, or 16 memories with 2048 total units (so, the size of each memory was $\frac{2048}{\text{num\_mems}}$), 1, 8, or 16 heads, 1 block of attention, and both the 'unit' and 'memory' gating methods. Figure 4 shows the results of a hyperparameter sweep scaled according to wall-clock time (models with more but smaller memories are faster to run than those with fewer but larger memories, and we chose to compare models with equivalent number of total units in the memory matrix $M$).

## A.2 Program Evaluation

To further study the effect of relational structure on working memory and symbolic representation we turned to a set of problems that provided insights into the RMC's fitness as a generalized computational model. The *Learning to Execute* (**LTE**) dataset [25] provided a good starting point for assessing the power of our model over this class of problems. Sample problems are of the form of linear time, constant memory, mini-programs.

Training samples were generated in batches of 128 on-the-fly. Each model was trained for 200K iterations using an Adam optimiser and learning rate of $1e^{-3}$. The samples were parameterized by literal length and nesting depth which define the length of terminal values in the program snippets and the level of program operation nesting. Within each batch the literal length and nesting value was sampled uniformly up to the maximum value for each - this is consistent with the *Mix* curriculum strategy from [25]. We evaluated the model against a batch of 12800 samples using the maximum nesting and literal length values for all samples and report the top score. Examples of samples for each task can be found in figure 6 and figure 7. It also worth noting that the modulus operation was applied to *addition*, *control*, and *full program* samples so as to bound the output to the maximum literal length in case of longer for-loops.

The sequential model consists of an encoder and a decoder which each take the form of a recurrent neural network [45, 25]. Once the encoder has processed the input sequence the state of the encoder is used to initialize the decoder state and subsequently to generate the target sequence (program output). The output from all models is passed through a 4-layer MLP - all layers have size 256 with an output ReLU - to generate an output embedding at each step of the output sequence.

In [25] teacher forcing is used for both training and testing in the decode phase. For our experiments, we began by exploring teacher forcing during training but used model predictions from the previous step as input to the the decoder at the next step when evaluating the model [45]. We also considered the potential effect of limiting

Figure 4: $N^{th}$ **Farthest hyperparameter analysis**. Timestamp refers to hours of training. There is a clear effect with the number of memories, with 8 or 16 memories being better than 1. Interestingly, when the model had 1 memory we observed an effect with the number of heads, with more heads (8 or 16) being better than one, possibly indicating that the RMC can learn to compartmentalise and relate information across heads in addition to across memories.

Figure 5: **LSTM and DNC training curves for the** $N^{th}$ **Farthest task**.

the dependency on the ground truth altogether when training the decoder [39] and using a non-auto-regressive regime where model predictions only were used during training. It turned out that this approach tended to yield the strongest results.

Following are the encoder/decoder configurations for a collection of memory models that performed best over all tasks. With the RMC we swept over two and four memories, and two and four attention heads, a total memory size of 1024 and 2048 (divided across memories), a single pass of self attention per step and scalar memory gating. For the baselines, the LSTM is a two layer model and we swept over models with 1024 and 2048 units per layer, skip connections and layer-wise outputs concatenated on the final layer. The DNC used a memory size of 80, word size 64, four read heads and one write head, a 2-layer controller sweeping over 128, 256 and 512 latent units per layer, larger settings than this tended to hurt performance. Also for the DNC, an LSTM controller is used for Program Evaluation problems, and feed-forward controller for memorization. Finally, the EntNet was compared with a total memory size of either 1024 or 2048 with 2, 4, 6, or 8 memory cells where total memory size is divided among memories and the states of the cells are summed to produce an output. All results reported are from the strongest performing hyper-parameter setting for the given model.

Addition (nesting = 2, literal length = 7):

```
x=473278230+(1257657+32721978)
print(x % 10**length)
A: 7257865
```

Control (nesting = 3, literal length = 3):

```
x = 221 if ((411 if 918 > 314 else 680) + 321) < 778 else 652
print(x % 10**length)
A: 221
```

Full Program (nesting = 2, literal length = 5):

```
x=82930-31249              x = (28694 if 89425 > 31990 else 38662)    x = (
for[6]                     for[5]                                        x=76957
   x+=98315                   x += 54926                                  for[7]
print(x % 10**length)      print(x % 10**length)                            x += 62117)
A: 641571                  A: 303324                                   for[8]
                                                                          x+=90285
                                                                       print(x % 10**length)
                                                                       A: 234056
```

Figure 6: Samples of *programmatic* tasks. Note that training samples will sample literal length up to including the maximum length.

**Copy:** $x_1 x_2 x_3 \ldots x_n$ $\Longrightarrow$ $x_1 x_2 x_3 \ldots x_n$

**Reverse:** $x_1 x_2 x_3 \ldots x_n$ $\Longrightarrow$ $x_n x_{n-1} x_{n-2} \ldots x_1$

**Double:** $x_1 x_2 x_3 \ldots$ $\Longrightarrow$ $x_1 x_2 x_3 \ldots x_n x_1 x_2 x_3 \ldots x_n$

Figure 7: *Memorization* tasks. Each sub-task takes the form of a list permutation.

As seen in figure 8 the RMC tends to quickly achieve high performance relative to the baselines, this demonstrates good data efficiency for these tasks especially when compared to the LSTM. From the same figure and table 1 (the results in the table depict converged accuracy scores for nesting 2 and literal length 5) it is also clear that the RMC scores well among the full set of program evaluation tasks where the DNC faltered on the *control* task and the EntNet on *copy* and *double* tasks. It should finally be noted that due to the RMC model size scaling with respect to total memory size over number of memories and consequently the top performing LSTM models contained many more parameters than the top performing RMC models.

## A.3 Viewport BoxWorld

We study a variant of BoxWorld, which is a pixel-based, highly combinatorial reinforcement learning environment that demands relational reasoning-based planning, initially developed in [46]. It consists of a grid of $14 \times 14$ pixels: grey pixels denote the background, lone colored pixels are keys that can be picked up, and duples of colored pixels are locks and keys, where the right pixel of the duple denotes the color of the lock (and hence the color of the key that is needed to open the lock), and the left pixel denotes the color of the key that would be obtained should the agent open the lock. The agent is denoted by a dark grey pixel, and has four actions: *up*, *down*, *left*, *right*. To make this task demand relational reasoning in a memory space, the agent only has perceptual access to a $5 \times 5$ RGB window, or viewport, appended with an extra frame denoting the color of the

Figure 8: *Programmatic* results. From left to right: *full program*, *addition*, *control*. The top row depicts per character accuracy scores from tasks with nesting = 2 and literal length = 5 while the bottom row shows scores from more difficult tasks with nesting = 3 and literal length = 6.

Figure 9: **Example BoxWorld level**. The left panel shows the full-view frame of a BoxWorld level. The agent, the dark grey pixel, only has access to a $5 \times 5$ view surrounding it (light gray area). The right panel shows the underlying graph that was sampled to generate the level. In this example the solution path has length 5 and there are 4 distractor branches.

key currently in possession. The goal of the task is to navigate the space, observe the key-lock combinations, and then choose the correct key-lock sequence so as to eventually receive the rewarded gem, denoted by a white pixel.

In each level there is a unique sequence of keys-lock pairs that should be traversed to reach the gem. There are a few important factors that make this task difficult: First, keys disappear once they are used. Since we include 'distractor' branches (i.e., key lock paths that lead to a dead end), the agent must be able to look ahead, and reason about the appropriate path forward to the gem so as to not get stuck. Second, the location of the keys and locks are randomised, making this task completely devoid of any spatial biases. This emphasises a capacity to reason about the relations between keys and locks, in memory, based on their abstract relations, rather than based on their spatial positions. For this reason we suspect that CNN-based approaches may struggle, since their inductive biases are tied to relating things proximal in space.

To collect a locked key the agent must be in possession of the matching key color (only one key can be held at a time) and walk over the lock, after which the lock disappears. Only then is it possible for the agent to pick up the adjacent key. Each level was procedurally generated, constrained to have only one unique sequence in each level ending with the white gem. To generate the level we first sampled a random graph (tree) that defined the possible paths that could be traversed, including distractor paths. An example path is shown in figure 9.

Figure 10: **Mini Pacman Results**.

We used a total of 20 keys and 20 locks (i.e., colors) in our sampling pool to produce each level. Three main factors determined the difficulty of the level: (1) the path length (i.e., number of locks) to the gem; (2) the number of distractor branches; and (3) the path lengths of the distractor branches. For training we used solution path lengths of at least 1 and up to 5, ensuring that an untrained agent would have a small probability of reaching the goal by chance, at least on the easier levels. We sampled the number of distractor branches to be between 0 and 5, with a length of 1.

The viewport observation was processed through two convolutional layers, with 12 and 24 kernels, and with $2 \times 2$ kernel sizes and a stride of 1. Each layer used a ReLU non-linearity. We used two extra feature maps to tag the convolutional output with absolute spatial position ($x$ and $y$) of each pixel/cell, with the tags comprising evenly spaced values between $-1$ and $1$. The resulting stack was then passed to the RMC, containing four memories, four heads, a total memory size of 1024 (divided across heads and memories), a single pass of self attention per step and scalar memory gating. For the baseline, we replaced the RMC with a $5 \times 5$ ConvLSTM with 64 output channels, with $2 \times 2$ kernels and stride of 1.

We used this architecture in an actor-critic set-up, using the distributed Importance Weighted Actor-Learner Architecture [47]. The agent consists of 100 actors, which generate trajectories of experience, and one learner, which directly learns a policy $\pi$ and a baseline function $V$, using the actors' experiences. The model updates were performed on GPU using mini-batches of 32 trajectories provided by the actors via a queue. The agent had an entropy cost of 0.005, discount ($\gamma$) of 0.99 and unroll length of 40 steps. The learning rate was tuned, taking values between $1e-5$ and $2e-4$. Informally, we note that we could replicate these results using an A3C setup, though training took longer.

The agent received a reward of $+10$ for collecting the gem, $+1$ for opening a box in the solution path and $-1$ for opening a distractor box. The level was terminated immediately after collecting the gem or opening a distractor box.

### A.3.1 Results

We trained an Importance Weighted Actor-Learner Architectures agent augmented with the RMC on BoxWorld levels that required opening at least 1 and up to 5 boxes. The number of distractor branches was randomly sampled from 0 to 5. This agent achieved high performance in the task, correctly solving 98% of the levels after $1e9$ steps. The same agent augmented instead with a ConvLSTM performed significantly worse, reaching only 73%.

### A.4 Language Modeling

We trained the Recurrent Memory Core with Adam, using a learning rate of 0.001 and gradients were clipped to have a maximum L2 norm of 0.1. Backpropagation-through-time was truncated to a window-length of 100. The model was trained with 6 Nvidia Tesla P100 GPUs synchronously. Each GPU trained with a batch of 64 and so the total batch size was 384. We used 512 (with 0.5 dropout) as the word embedding sizes, and tied the word embedding matrix parameters to the output softmax.

We swept over the following model architecture parameters:

- Total units in memory $\{1000, 1500, 2000, 2500, 3000\}$
- Attention heads $\{1, 2, 3, 4, 5\}$

- Number of memories $\{1, 2\}$
- MLP layers $\{1, 2, 3, 4, 5\}$
- Attention blocks $\{1, 2, 3, 4\}$

and chose 2500 total units, 4 heads, 1 memory, a 5-layer MLP, and 1 attention block based upon validation error on WikiText-103. We used these same parameters for GigaWord and Project Gutenberg without additional sweeps, due to the expense of training.

Figure 11: **Validation perplexity on WikiText-103**. LSTM comparison from [32]. Visual display of data may not match numbers from table 2 because of curve smoothing.

Figure 12: **Perplexity as a function of test unroll length.** Increase in perplexity when models are unrolled for shorter sequence lengths at test time without state transfer between unrolls. Perplexities are compared against the 'best' perplexity where the model is unrolled continuously over the full test set. We see that both models incorporate little information beyond 500 words. Furthermore, the RMC has a smaller gain in perplexity (drop in performance) when unrolled over shorter time steps in comparison to the LSTM, e.g. a regression of 1 perplexity for the RMC vs 5 for the LSTM at 100 time steps. This suggests it is focusing on more recent words in the text.

Table 3: **Test perplexity split by word frequency on GigaWord v5.** Words are bucketed by the number of times they occur in training set, $> 10K$ contains the most frequent words.

|  | $> 10K$ | 10K-1K | $< 1K$ | All |
|---|---|---|---|---|
| LSTM [32] | 39.4 | 6.5e3 | 3.7e4 | 53.5 |
| LSTM + Hebbian Softmax [32] | 33.2 | 3.2e3 | **1.6e4** | 43.7 |
| RMC | **28.3** | **3.1e3** | 6.9e4 | **38.3** |