[Reviews · NeurIPS 2018]

Reviewer 1



This paper proposes relational recurrent neural networks to incorporate relational reasoning into memory-based neural networks. More specifically, the authors design a new memory module, called relational memory core (RMC), to make memories to interact. Multi-head dot product attention is used for this purpose. Experiments are constructed on a few supervised learning and reinforcement learning tasks, which may profit from more capable relational reasoning across sequential information, to show the effectiveness of the proposed method. The main strengths of this paper are as follows: (1) It is a novel and well motivated idea to design a new deep learning architecture which can perform relational reasoning. (2) The proposed network structure, RMC, is a reasonable memory module to allow the memories to interact. (3) The experiments are extensive. The results show explicit performance improvements. The main weaknesses of this paper are as follows: (1) Some claimed points are not very understandable to me. First, it is unclear how the intuition (that memory architectures involving relational reasoning benefit some learning tasks) can be confirmed. Second, how can you claim that the proposed method has the capacity of relation reasoning from the evidence of "improved function". These points are the footstones of this paper, which should be clearly analyzed, instead of simple explanation. (2) The analysis of (hyper)parameters should be analyzed in the experiments. It will be nice to show the influence of of the attention model with different settings. (3) The presentation can be improved. For example, Figure 3 are not clear enough for understanding. (4) Another paper titled "Relation Deep Reinforcement Learning" is closely related to this one, thus should be cited and discussed.

Reviewer 2



In this work, the authors introduce a Relational Memory Cell (RMC) which could be injected to LSTM to solve the relational reasoning task. Results on supervised learning tasks and reinforcement learning tasks show that the model outperforms standard LSTM and several baseline sequential models. Despite the motivation is attractive and the name of the model is fancy, I think this work actually just applies the self-attention to the LSTM memory cell, which I believe is very trivial and gives limited inspirations to the readers of this article.

Reviewer 3



The paper presents a memory-based neural network architecture for reasoning tasks. The authors focus on a variety of tasks of relational reasoning in the context of video-games (pacman), natural language, and algorithmics. As the paper is well written, the improvement looks marginal compared to SoA. However, my main concern is the absence of the MAC cell (Manning et al 2018) which have demonstrated already strong results in such relation reasoning task even if the modification of the LSTM cell in section 3.3 looks related. Finally, the multi-head interactive attention mechanism proposed in the paper is a marginal innovation regarding SoA of this domain that seems related to self attention mechanism which has been heavily used in the context of machine reading recently.